# Leg Configuration Analysis and Prototype Design of Biped Robot Based on Spring Mass Model

**Junjie Che [1], Yang Pan [2,\*], Wei Yan [1] and Jiexian Yu [1]**

1   Shenzhen Key Laboratory of Biomimetic Robotics and Intelligent Systems, Department of Mechanical and Energy Engineering, Southern University of Science and Technology, Shenzhen 518055, China; 11930360@mail.sustech.edu.cn (J.C.); 11930472@mail.sustech.edu.cn (W.Y.); yujx3@mail.sustech.edu.cn (J.Y.)

2   Guangdong Provincial Key Laboratory of Human-Augmentation and Rehabilitation Robotics in Universities, Southern University of Science and Technology, Shenzhen 518055, China

\*   Correspondence: pany@sustech.edu.cn

**Abstract:** The leg structure with high dynamic stability can make the bionic biped robot have the inherent conditions to perform elastic and highly dynamic motion. Compared with the quadruped robot, the leg structure of the biped robot is more complex and has more degrees of freedom. This also complicates kinematic and dynamic modeling. In this paper, the kinematics model of a bionic biped robot is established. The leg configuration of the robot is a series parallel hybrid mechanism with five active joints and six passive joints. The mechanism is a spring mass model that interacts organically with the environment and mimics the characteristics of human walking well. By analyzing the topological configuration of leg mechanism, we use the screw theory to establish the forward and inverse kinematics models. Then, we build the prototype, and use a step gait to test the model and prototype. The research of this paper has obvious application significance for the design and iteration of biped robot prototype.

**Keywords:** biped robot; kinematics; biped robot prototype; leg configuration

## 1. Introduction

A biped robot is a highly non-linear, multi-coupling mobile robot. Its high flexibility enables it to span a wide range of unstructured scenarios, which enables it to perform many complex and variable tasks instead of humans. This also requires a biped robot to perform fast and stable dynamic motion and have high adaptability to the environment. Therefore, the leg structure that has high dynamic stability has become a difficult point in the design of biped robots. For this reason, researchers have done a lot of meaningful work and developed several foot-based robots. Atlas developed by Boston Dynamics is the most dynamic biped robot in the world. It has 28 joints and weighs 80 kg [1]. It can not only complete tasks like jumping and running, but also perform high-difficulty continuous sports like running and dynamic dancing. The performance shown in video is undoubtedly excellent, but the hydraulic drive leg structure based on 3D printing has some drawbacks such as high cost, difficult to produce and control [2]. ASIMO, which has been able to walk dynamically since the beginning of the 20th century, is the result of many years of research on biped robots by Honda Corporation of Japan. It weighs 48 kg, has 12 degrees of freedom for both feet, and has a maximum moving speed of 9 km/s [3,4]. Sebastian Lohmeier and Thomas Buschmann designed a 25-degree-of-freedom humanoid walking robot LOLA to experiment with a fast, human-like walking motion. It has seven degrees of freedom in its legs, an overall weight of 55 kg, and a maximum speed of 5 km/h [5]. Other similar types of robots include Walkers from UBTech in China [6], HRP series from AIST in Japan [7,8], TORO from DLR in Germany [9,10], etc. The legs of these robots are basically fully articulated. This type of robot has the advantages of simple design and easy control, but rigid mechanical connections allow the joints of the robot to absorb the impact of rigid contact with the ground while moving. This is not conducive to the flexible, highly

dynamic motion of the robot similar to that of humans and animals [11]. Another type of biped robot is designed using passive dynamics, such as Delft's robots Flame and Tulip [12], passive dynamic walkers [13], LARMbot 2 [14], and Cornell Ranger, which travels 64 km on a single charge [15]. These types of robots can exhibit more efficient mobility without relying on driving or on a small amount of driving, but their movement is limited to specific gaits and environments [11].

At present, it is very rare for dynamic biped robots to have the ability to walk, run or jump at the same time in the world. Most biped robots are either multifunctional, static or dynamic, and limited to a specific gait [16]. The main reason is that the leg structure lacks natural compliance and cannot achieve the organic physical interaction between the natural biological legs and the environment [17]. Few robots can increase the energy efficiency of dynamic motion while enhancing its dynamic capability. Digit and Cassie, two-legged robots of Agility Robot, are successful examples in this regard [18]. Digit, an advanced version of Cassie, is a humanoid with both hands and legs. As the lower body of Digit, Cassie is an ostrich-like biped robot and a compliant actuated robot. It has 12 joints on its legs, including five active joints and two passive joints consisting of elastic [19]. During high-speed motion, this leg structure design can absorb the impact of the end-to-ground contact and is conducive to exhibiting flexible and highly dynamic motion. At the same time, a nearly fully hinged leg structure can make motion control more precise. The addition of a four-bar mechanism reduces the inertia of the end-to-end of the robot, thereby improving the performance of the robot and reducing Load of the motor. While reading the relevant literature, there was no literature on the kinematics analysis of Cassie, which focused more on the study of trajectory planning and control strategies [20–22].

The spring mass model can well mimic the human motion characteristics. As one of the representatives of humanoid robots, compliant actuated bipedal robot has better dynamic performance than other types of robots. In this paper, a kinematics model and a primary prototype of leg structure based on spring mass model are built. The main difficulty is that the flexible element makes the control of the prototype difficult. In addition, the organic combination of the flexible element and the prototype is one of the difficulties in the design of the prototype. In order to facilitate the further research and experimental platform of compliant actuated bipedal robot, we focus on this compliant actuated bipedal robot's leg topology is used for kinematics analysis, prototype building and preliminary performance testing. The Section 2 mainly carries out the kinematic analysis of the leg configuration. The leg configuration of the experimental platform is described. The positive kinematics of the mechanism and the inverse kinematics of the leg structure are solved using the screw theory, and the corresponding validation and simulation are carried out. The specific data and construction of the experimental prototype will be shown in the Section 3. The Section 4 explains the gait of the robot, the experimental process and the corresponding results.

## 2. Kinamatic Models

This section is divided into four main sections. Their contents are to analyze the leg configuration of the prototype, and on this basis, to derive and verify the forward and inverse kinematics solutions. This chapter carries out more detailed kinematics analysis, forward and backward decomposition derivation and gait verification for the optimized leg configuration of the prototype in the previous work [23].

### 2.1. Configuration of the Legs of the Biped Robot

The leg structure studied in this paper is a hybrid structure based on the spring mass model. In the body coordinate system of the robot, the front of the robot is in the X-axis direction. The vertical direction is in the Y-axis direction. The Z-axis points to the right side of the robot. Figure 1 shows the XOZ plane of the robot with the Z-axis vertical paper facing out. Figure 2 show the side view and mechanism diagram of biped robot. The structure sketch shows that there are 12 joints and 5 degrees of freedom in the legs of

the robot, among which $q_1$, $q_2$, $q_3$, $q_4$ and $q_5$ are active joints. The active joints $q_1$, $q_2$ and $q_3$ mainly simulate the movement of the human hip, i.e., along the $XYZ$ direction of the leg coordinate system. The knee movement is mainly controlled by the active joint $q_4$, which controls the lifting and lowering of the calf. The active joint $q_5$ controls the movement of the ankle, but only with a single degree of freedom. It is worth noting that the active joint $q_4$ controls the parallel part of the entire leg structure. Its rotation causes the entire leg to undergo some degree of displacement and rotation along the joint $C$. In addition, the position of the leaf spring is installed at the rod $BO$ and the rod $DN$ in the figure. Because spring elastic deformation occurs only when it is subjected to a relatively large impact in dynamic motion, dynamic analysis is required. In kinematics analysis, the spring is assumed to be a rigid body. This greatly reduces the difficulty of solving kinematics. To facilitate calculation, it is simplified. The dashed line part in the diagram is the actual structure sketch for kinematics analysis.

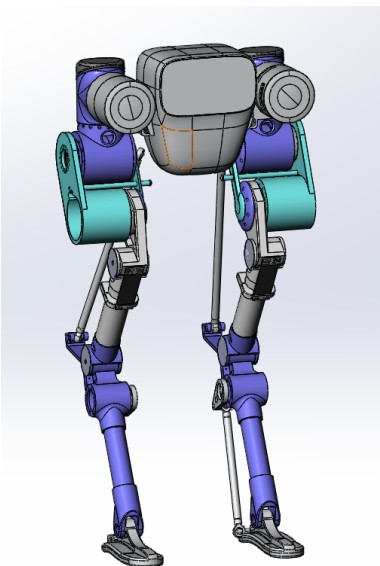

**Figure 1.** Simulation model of biped robot.

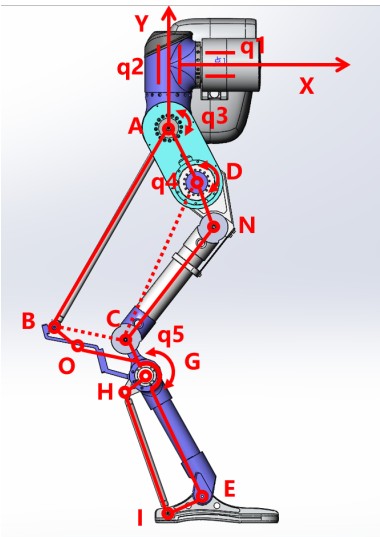

**Figure 2.** Side view and mechanism diagram of biped robot simulation model.

### 2.2. Forward Kinematic Models

Because the leg configuration of the robot is a hybrid configuration, the parallel part of the hybrid mechanism needs to be split properly in the process of solving the positive kinematics. When only the active joint $q_4$ is rotated, the quadrilateral $ABCD$ is distorted. A certain degree of displacement and rotation of the whole leg changes the position and posture of the end. Obviously, the pose matrix of this part of the mechanism can not be simply calculated. But when rod $AB$ and rod $BC$ are ignored, joint $C$ is regarded as an active joint the rod, the whole leg can be considered as a series connection. The tstructure is used to solve the positive kinematics. According to the spinor theory, the pose matrix of the corresponding joint can be obtained only by knowing the position and rotation direction of each active joint. Finally, multiplying them in turn can solve the end position of the joint rotation of the active joint.

$$R = E \cos\theta + \sin\theta\vec{\omega} \times + (1 - \cos\theta)\vec{\omega} \times \vec{\omega} \times \tag{1}$$

$$\vec{p} = (E - R)\vec{\omega} \times \vec{v} + \theta(\vec{\omega} \cdot \vec{v})\vec{\omega} \tag{2}$$

$$P = {}^0_1T \, {}^1_2T \, {}^2_3T \, {}^3_4T \, {}^4_CT \, {}^C_ET \, P_0 \tag{3}$$

where, $P$ is the end position. $P_0$ is the initial position of the end. ${}^0_1T, {}^1_2T, {}^2_3T, {}^3_4T, {}^4_CT, {}^C_ET$ is the pose matrix of each active joint. The rotation angle of the pose matrix of joint $E$ can be calculated by $q_5$.

### 2.3. Inverse Kinematics

The purpose of solving inverse kinematics is to calculate the rotation of the active joint corresponding to the end displacement when the robot moves, so that the robot can accurately follow the prescribed trajectory. The leg configuration studied in this paper has the following characteristics. First, all the joints below the crown are always on the same plane. Then, the active joints $q_1$ and $q_2$ of the robot buttock control the rotation posture of the leg plane. Finally, the rod $OA$ is always on the plane $YOZ$. Therefore, the idea of solving inverse kinematics in this chapter is to find out the coordinates of joint $E$ in the leg coordinate system using known conditions such as position and posture of the end, and then convert it into the coordinates of the leg plane, so as to convert a three-dimensional space problem into a two-dimensional space problem, and finally, to gradually calculate the rotation of all active joints according to the structure.

The rod length, end coordinates $M(x_0, y_0, z_0)$, the position of the end at the sole of the foot $IM$ (from the heel position) and the toe pointing $\vec{n}(a, b, c)$ are known. The main idea is to first find the line of intersection between the leg plane and the plane $YOZ$ on the plane $YOZ$ according to the toe pointing and the end coordinate, and then $q_1$ can be obtained according to the slope of the line. On the plane $YOZ$, the left and right sides of the intersection line are judged at the end, and then $q_2$ can be obtained. By transforming the end coordinate and the toe direction into the coordinates and vectors of the leg plane, the coordinates of the joint $E$ can be obtained, and the coordinates of the joint $E$ can be used. $q_3$ and $q_4$ can be obtained from the length of each rod. Finally, $q_5$ can be obtained from the coordinates of the tips and the ends.

#### 2.3.1. Solving $q_1$

First, the leg plane equation is derived from the end coordinates and the tip pointing.

$$\overrightarrow{OM} = (x_0, y_0, z_0) \tag{4}$$

$$\vec{m} = \overrightarrow{OM} \times \vec{n} = (x_m, y_m, z_m) \tag{5}$$

$$x_a \, x_m + y_a \, y_m + z_a \, z_m = 0 \tag{6}$$

To find the line of intersection between the leg plane and the plane $YOZ$, replace $X_a = 0$ into the Equation (1), and get the line of intersection equation.

$$y = -\frac{z_m}{y_m} z \tag{7}$$

$$k = -\frac{z_m}{y_m} \tag{8}$$

In this case, you can find $q_1$. There are four cases of $q_1$, $k > 0, k = 0, k < 0, k \to \infty$. Make $\theta_1$ that the angle between the above intersection line and the $Z$-axis.

$$q_1 = \begin{cases} \frac{\pi}{2} - \theta_1, \theta_1 \in \left(0, \frac{\pi}{2}\right) \\ -\frac{\pi}{2}, \theta_1 = 0, z > 0 \\ \frac{\pi}{2}, \theta_1 = 0, z < 0 \\ -\theta_1 - \frac{\pi}{2}, \theta_1 \in \left(-\frac{\pi}{2}, 0\right) \\ 0, \theta_1 = \frac{\pi}{2} \text{ or } -\frac{\pi}{2} \end{cases} \tag{9}$$

2.3.2. Solving $q_2$

Before finding $q_2$, judge whether the end is on the left or right of the intersection line on the plane $YOZ$.

When $\theta_1 = \frac{\pi}{2}$ or $-\frac{\pi}{2}$,

$$\begin{cases} left, z > 0 \\ right, z < 0 \\ q2 = 0, z = 0 \end{cases} \tag{10}$$

When $\theta_1 \in \left(0, \frac{\pi}{2}\right)$,

$$\begin{cases} left, y + \frac{z_m}{y_m} < 0 \\ right, y + \frac{z_m}{y_m} > 0 \\ q2 = 0, y + \frac{z_m}{y_m} = 0 \end{cases} \tag{11}$$

When $\theta_1 = 0$

$$\begin{cases} left, \left(q_1 = \frac{\pi}{2}, y < 0\right) \text{ or } \left(q_1 = -\frac{\pi}{2}, y > 0\right) \\ right, \left(q_1 = \frac{\pi}{2}, y > 0\right) \text{ or } \left(q_1 = -\frac{\pi}{2}, y < 0\right) \\ q2 = 0, \left(q_1 = \frac{\pi}{2}, q_2 = 0\right) \text{ or } \left(q_1 = -\frac{\pi}{2}, q_2 = 0\right) \end{cases} \tag{12}$$

When $\theta_1 \in \left(-\frac{\pi}{2}, 0\right)$,

$$\begin{cases} left, y + \frac{z_m}{y_m} > 0 \\ right, y + \frac{z_m}{y_m} < 0 \\ q2 = 0, y + \frac{z_m}{y_m} = 0 \end{cases} \tag{13}$$

Then by judging left and right, $q_2$ is calculated in different ways, $\theta_2$ is the angle between $NO$ projected by $MO$ on the plane $YOZ$ and $Z$-axis. When the end is on the right side, there are many cases depending on $\theta_1$ value.

$$\begin{cases} \angle NOP = \theta_1 - \theta_2, \theta_1 \in \left(0, \frac{\pi}{2}\right) \\ \angle NOP = \theta_2, \theta_1 = 0 \\ \angle NOP = \theta_1 + \theta_2, \theta_1 \in \left(-\frac{\pi}{2}, 0\right), z > 0 \\ \angle NOP = \frac{\pi}{2} + \theta_1, \theta_1 \in \left(-\frac{\pi}{2}, 0\right), z = 0 \\ \angle NOP = \pi + \theta_1 - \theta_2, \theta_1 \in \left(-\frac{\pi}{2}, 0\right), z < 0 \\ \angle NOP = \frac{\pi}{2} - \theta_2, \theta_1 = \frac{\pi}{2} \end{cases} \tag{14}$$

$$NP = NO \ \sin \angle NOP \tag{15}$$

$$\angle MPN = \arctan \frac{MN}{NP} \tag{16}$$

Based on the values, $q_2$ has several values below.

$$\begin{cases} q_2 = \frac{\pi}{2} - \angle MPN, x > 0 \\ q_2 = -\frac{\pi}{2} + \angle MPN, x < 0 \\ q_2 = \frac{\pi}{2}, x = 0, c > 0 \\ q_2 = -\frac{\pi}{2}, x = 0, c < 0 \end{cases} \tag{17}$$

When the end is on the left, the calculation method changes slightly, but the rationale is basically the same. The $q_2$ values are as follows.

$$\begin{cases} q_2 = -\frac{\pi}{2} + \angle MPN, x > 0 \\ q_2 = \frac{\pi}{2} - \angle MPN, x < 0 \\ q_2 = -\frac{\pi}{2}, x = 0, c > 0 \\ q_2 = \frac{\pi}{2}, x = 0, c < 0 \end{cases} \tag{18}$$

### 2.3.3. Coordinate Conversion

To reduce the complexity of calculating the inverse solution, the end is transformed into the corresponding coordinate in the leg plane coordinate system. The foot end geometry is showed by Figure 3. Set the leg plane formed by rotating $ABCD$ on the leg to be a plane $XOY$, point $O$ to be the same as the original coordinate, and point $O$ to be the same as the $X$-axis. When $q_1$ is 0, the $Y_0$ axis is the same as the $Y$-axis, and when $q_1$ is not 0, the $Y_0$ axis is the new axis after rotating $q_1$.

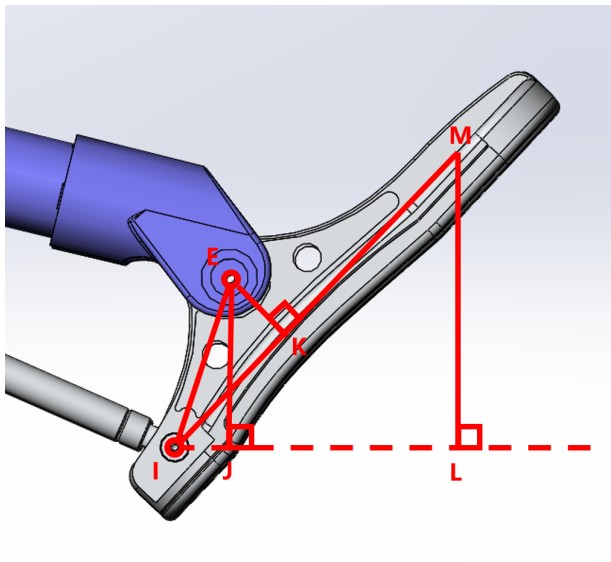

**Figure 3.** Foot end geometry.

When $q_2 = 0$,

$$x_0 = x \tag{19}$$

$$y_0 = -NO \tag{20}$$

When $q_2 \neq 0$,

$$x_0 = \frac{MN}{\sin \angle MPN} \frac{x}{|x|} \tag{21}$$

$$y_0 = -\sqrt{NO^2 - NP^2} \tag{22}$$

Then the coordinates $(x, y)$ of the joint $E$ are calculated, as shown in the figure. Horizontal lines are made along the joint $I$, and the intersections of the joint $E$ and the end $M$ to the horizontal line are points $K$ and $L$. The direction of the foot is determined by determining whether the end point is above or below the coordinate $XOY$ and the size of $\angle EIL$. Based on the geometric relationship, the coordinates of the joint $E$ are calculated as follows:

$$\begin{cases} \begin{cases} x_1 = x_0 - IK - IL \\ y_1 = y_0 - LM + EK \end{cases} , -k\,c < b, \angle EIK + \angle LIM > \frac{\pi}{2} \\ \begin{cases} x_1 = x_0 - IL \\ y_1 = y_0 - LM + EK \end{cases} , -k\,c < b, \angle EIK + \angle LIM = \frac{\pi}{2} \\ \begin{cases} x_1 = x_0 + IK - IL \\ y_1 = y_0 - LM + EK \end{cases} , -k\,c < b, \angle EIK + \angle LIM < \frac{\pi}{2} \\ \begin{cases} x_1 = x_0 - IM + IJ \\ y_1 = y_0 + EJ \end{cases} , -k\,c = b \\ \begin{cases} x_1 = x_0 + IK - IL \\ y_1 = y_0 + LM + EK \end{cases} , -k\,c > b \end{cases} \tag{23}$$

where,

$$\angle LIM = \arctan \frac{\sqrt{b^2 + c^2 + \left(\frac{a}{\cos q_2}\right)^2 - a^2}}{\frac{a}{\cos q_2}} \tag{24}$$

$$\angle EIJ = \arctan \frac{EJ}{IJ} \tag{25}$$

### 2.3.4. Solving $q_3$ and $q_4$

When calculating $q_3$, it is important to note that joint $E$ is solved slightly differently on the $Y_0$ axis or on the left and right ends, so a simple classification is needed. Set the projection of joint $E$ on the $Y_0$ axis of the leg plane as point $F$, and $\angle DAO$ as $q_{30}$ in the initial state. Based on the coordinates of joint $E$ on the leg plane and the rods, use the trigonometric function to calculate the values of each angle and to determine the position of the end.

$$\begin{cases} \angle DAF = \angle BAD - \angle BAE + \angle EAF, x_1 \geqslant 0 \\ \angle DAF = \angle BAD - \angle BAE - \angle EAF, x_1 < 0 \end{cases} \tag{26}$$

$q_{30}$ can be requested by the above methods.

$$q_3 = \pi - \angle DAF - q_{30} \tag{27}$$

$q_4$ is also solved by trigonometric functions.

$$q_4 = \angle ADC - \frac{3}{4}\pi \tag{28}$$

### 2.3.5. Solving $q_5$

We set the angle of $CE$ intersecting the horizontal line at point $E$, forming two angles with the horizontal line, the angle of negative $x_0$-axis is $\theta_3$, the angle of negative $x_0$-axis is $\theta_4$, and the angle of negative $x_0$-axis is . Using and , is obtained, then is obtained by trigonometric function, and then is obtained. The solution of and requires situational judgment.

$CE$ intersect the horizontal line at point $E$ and form two angles with the horizontal line. The angle of negative $x_0$-axis is $\theta_3$. In the same time, $EI$ form two angles with the horizontal line. The angle of negative $x_0$-axis is $\theta_4$. Using $\theta_3$ and $\theta_4$, $\angle GEI$ is obtained, then

$\angle EGH$ is obtained by trigonometric function, and then $q_5$ is obtained. The solution of $\theta_3$ and $\theta_4$ requires situational judgment.

$$\begin{cases} \theta_3 = \angle AEF + \angle BEC - \angle AEB, x_1 > 0 \\ \theta_3 = \frac{\pi}{2} + \angle BEC - \angle AEB, x_1 = 0 \\ \theta_3 = \pi - \angle AEF + \angle BEC - \angle AEB, x_1 < 0 \end{cases} \tag{29}$$

$$\begin{cases} \theta_4 = \angle EIK, -k\ c = b \\ \theta_4 = \angle EIJ + \angle LIM, -k\ c < b \\ \theta_4 = \angle EIJ - \angle LIM, -k\ c > b \end{cases} \tag{30}$$

$$\angle GEI = \theta_3 + \theta_4 \tag{31}$$

Since the control mechanism of the sole of the foot is a four-bar mechanism, there are maximum and minimum values for $\angle GEI$. When the rod $GH$ and the rod $HI$ are in the same straight line, $\angle GEI$ is maximum, and when the rod $HI$ and the rod $EI$ are in the same straight line, $\angle GEI$ is the smallest at this time. When $\angle GEI$ value is greater than the maximum value, the maximum value is taken, which is the same as the minimum value. The two extremes can be calculated by a trigonometric function. When $q_{50}$ is the initial state, $\angle EGH$ is 0. The $q_5$ can be obtained from the geometric relationship.

$$q_5 = \angle EGH - q_{50} \tag{32}$$

### 2.4. Verification of Kinematic Model

Kinematics is mainly validated by one-legged simulation. In order to better show the correctness of the inverse kinematics, the walking gait of the robot is simulated. The principle is that the center of gravity is always kept in the support polygon during the walking of the robot to prevent the robot from falling. Because there is a gap between the simulation model and the actual prototype, this part is only used to verify some gait and kinematics models. On the other hand, the end is in face contact with the ground, but the end joint has only one degree of freedom. This greatly limits the gait selection of the robot. There are two ways to solve this problem. One is to change the contact surface between the end and the ground to point contact or line contact. Another way is to add degrees of freedom to the end. These problems will be solved in the development of the next generation prototype. Figure 4 is a simulation of the stable walking of the robot.

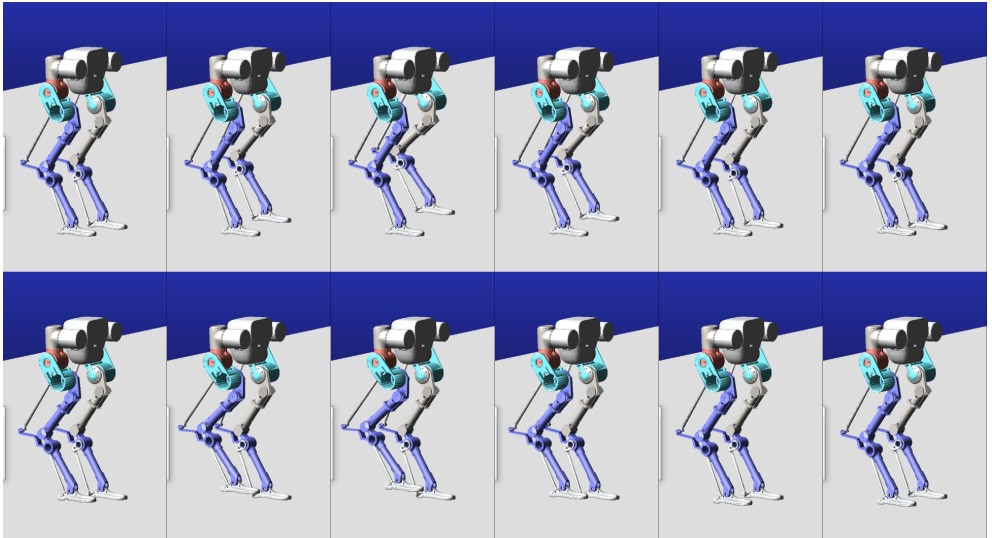

**Figure 4.** Step simulation. In the simulation, the robot steps at a speed of 4 mm/ms. The above figure shows two steps in intercepting the robot's motion.

## 3. Experimental Prototype and Its Construction

The prototype used in this paper is the first-generation prototype, showed in Figure 5. The first-generation prototype is designed independently on the basis of Cassie leg structure combined with the size and weight of the driving module. The design tends to test the force and motion state of the leg structure. Therefore, both the power supply and the controller are designed with an external connection. It is hoped that the design experience of the first generation will provide more valuable optimization for the design of the next generation.

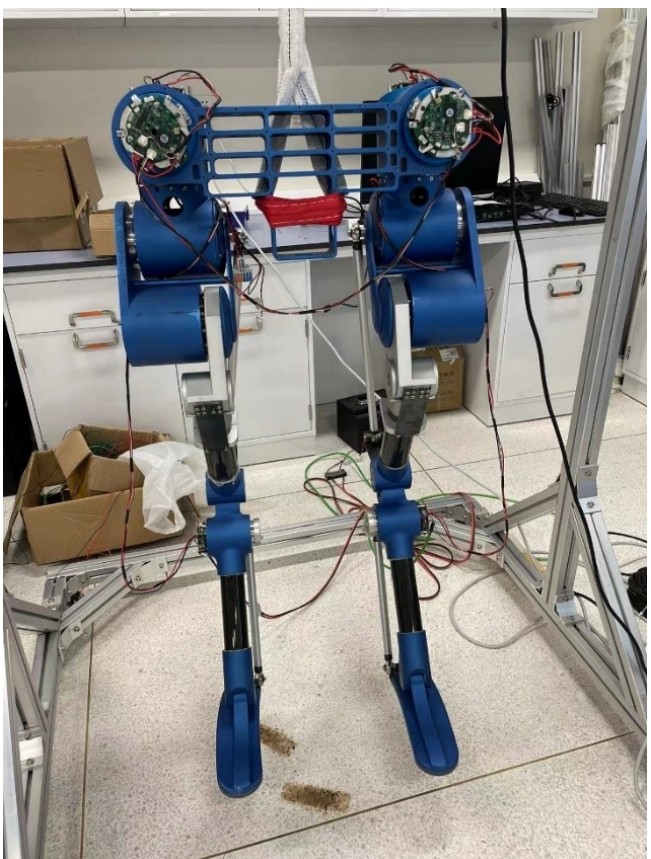

**Figure 5.** Experimental Prototype.

### 3.1. Structural Component Design

For foot-type robots, especially biped robots, weight has a great impact on their performance. Therefore, the mechanical structure should be as lightweight as possible. High strength and stiffness are also essential features to prevent serious deformation of the robot during motion. In the design process, the material selection and structure design of most parts are done through experience and intuition. Figure 6 show the side view of experimental prototype. Aluminum alloy is used in the motor housing and all kinds of connectors. Aluminum alloy has the advantages of low quality, high strength and low price. Tube carbon fibers are used in the middle of lower leg to reduce the end weight. The use of carbon fibers reduces the weight at the end of leg and the inertia of end rotator. The knee joint part is a four-bar mechanism, showed in Figure 7. By turning the active joint $q_4$, the four-bar mechanism produces a distortion that lifts the lower leg, thus enabling the knee to bend. Properly increasing the size of the rod BC can improve the performance of the active joint $q_4$. Similarly, shortening the rod $GH$ or the rod $EI$ can reduce the load of the active joint $q_5$.

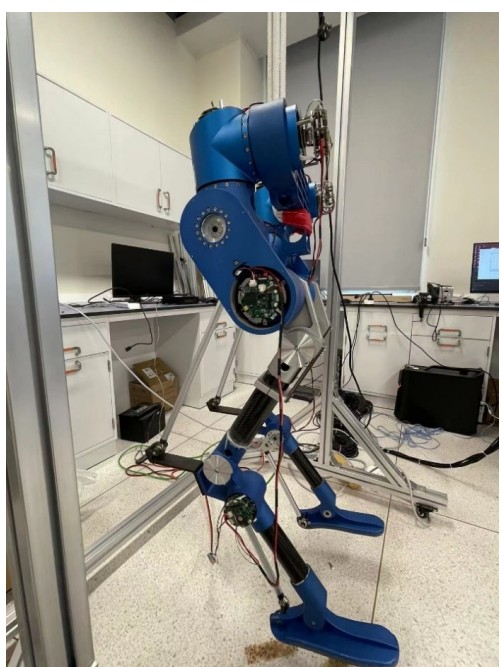

**Figure 6.** Side view of experimental prototype.

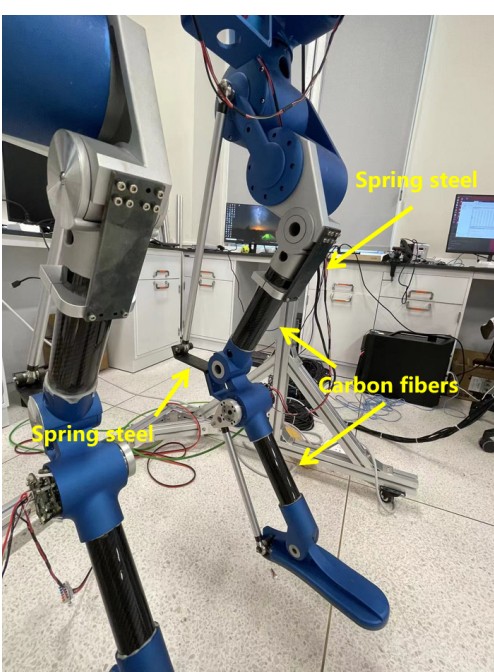

**Figure 7.** Inside view of experimental prototype.

### 3.2. Driver Design

The development of compact and lightweight transmission systems is critical for biped robots. This not only reduces overall weight, but also makes the body more compact and reduces overall control difficulty. Therefore, drivers use an integrated module to save space and weight to joints. Composition of motor module is showed in Figure 8. In the module, the motor uses a brushless DC frameless moment motor. Because they have the advantages of low weight, low rotational inertia, fast dynamic response and high accuracy. The module uses a dual encoder design, which enables precise joint control and robot dynamic development. The decelerator uses a harmonic decelerator.

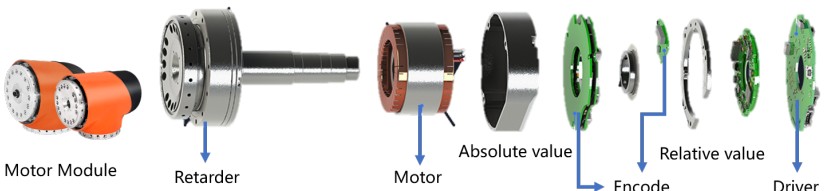

**Figure 8.** Composition of motor module.

*3.3. Control*

The control of the robot is mainly accomplished by a self-designed controller in our laboratory. The controller is a real-time controller based on a Linux system. Its real-time performance is mainly accomplished by EtherCAT, an industrial Ethernet bus technology, and Xenomai, a strong real-time extension of the Linux kernel. EtherCAT enables the controller to transmit real-time short frame data, enabling real-time communication. Xenomai enables real-time instructions to run at the highest priority of the system, ensuring the real-time performance of the system. The combination of the two achieves millisecond-level motion control, which enables the robot to execute more efficiently. At the same time, real-time also allows the driver to instantly feedback the data of the prototype motion, which makes the robot motion more controllable.

**4. Prototype Performance Test**

The difficulty of biped robot prototype is how to balance the organic relationship between motor performance and weight. In order to enable the robot to meet the performance requirements of dynamic motion, during the assembly and debugging phase of the prototype, some parts of the prototype are optimized to make the robot have better motion performance. For example, the size of the closed chain part is modified to greatly improve the bearing capacity of the motor. After a series of performance optimization, a simple gait experiment is carried out on the prototype. The experiment is mainly divided into suspension step and ground step. In both experiments, the trajectory planning of the robot is the same. Air walking is mainly to verify whether the robot can perform well. The following subsections mainly describe the trajectory planning of the robot used in the prototype performance test and the results of some experiments.

*4.1. Experiment*

The experiment is divided into two parts. Experiment A is an experiment of walking in the air. Experiment B is an experiment of walking on flat ground. The first generation of biped robots were very large and weighed nearly 60 kg. The motor used is also an industrial-grade module. Furthermore, biped robots differ from mechanical arms in that they do not have a base to hold the robot. Therefore, how to ensure the safety of the robot itself is very important during the experiment. As shown in the Figure 9, the prototype of the robot is hoisted by a gantry with lifting function. Because the prototype is too heavy and can easily damage the structure and motor of the robot when it falls, the main function of the gantry is to protect the robot, that is, to hang the robot when it is about to fall. When a robot walks in the air, it hangs in the air. When walking on the ground, the robot is also suspended by a rack. At this point, the robot is standing on the ground with both feet. During the course of the motion, the attached rope only works when the body of the robot tilts more heavily.

The main purpose of this experiment is to test whether the motor can meet the dynamic motion performance requirements of biped robot. Because there is no external awareness device such as IMU, the robot steps in a blind way. In the trajectory planning of a robot, the body trajectory is the direction of movement in the world coordinate system. When the robot takes a step, the trajectory at the end is semielliptical. The velocity curve is a T-shaped curve. The trajectory planning for the robot walking is shown in the figure. Since the robot's gait was a step during the experiment, the forward distance was 0. The end

result is that your body stays the same and your legs step upright. By observing the motion of the prototype in the air and on the ground, you can know whether the robot is stable or not. Using the current data measured by the motor module, you can know whether the motor performance of the robot meets the requirements of the robot motion.

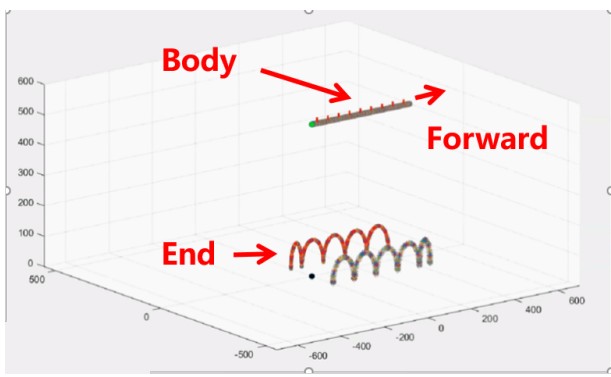

**Figure 9.** Trajectory planning of body and end.

### 4.2. Experiments Result

#### 4.2.1. Experiments A

The main purpose of Experiment A is to verify whether the robot can move according to the instructions and to test the maximum speed of the end movement of the robot. Experiment screenshot and data is showed in Figures 10 and 11. Thus, the available speed range of experiment B and the motion performance of the robot can be simply evaluated. The following is the current diagram corresponding to each motor collected. The data collected is the permille of current. Through the experimental data, verify that the motion trajectory of the robot is consistent with the simulation. At the same time, it can be know that when the average end speed is 41.7 mm/s (the uniform speed of T-shaped curve is 8 mm/ms), the current of the motor $q_4$ is close to the rated current or even exceeds the rated current for a short time. The main reason is that the motor $q_4$ motion range is large, the speed is fast, and the end weight is not low. Other motor current values are within the rated range. The motor $q_1$ does not move, but the current value reaches about 200 thousandths. The main reason is that the robot moves faster, which makes the robot shake as a whole. If the end weight is large, the motor $q_1$ needs to increase the torque input in order to maintain the original position. The motor $q_2$ also has no output, and it is normal that the current value is more than ten thousandths. Although the motor $q_3$ has a large load, its motion range is small. Its current value is within 400 thousandths. Through analysis, it can be seen that it's necessary to leave enough performance space for the robot because the collision between the robot and the ground will impact the motor to a certain extent.

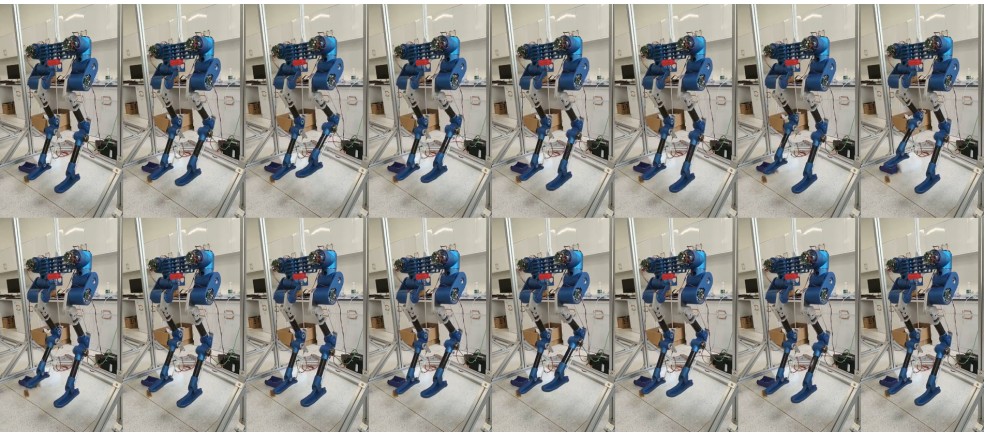

**Figure 10.** The prototype walks step in the air.

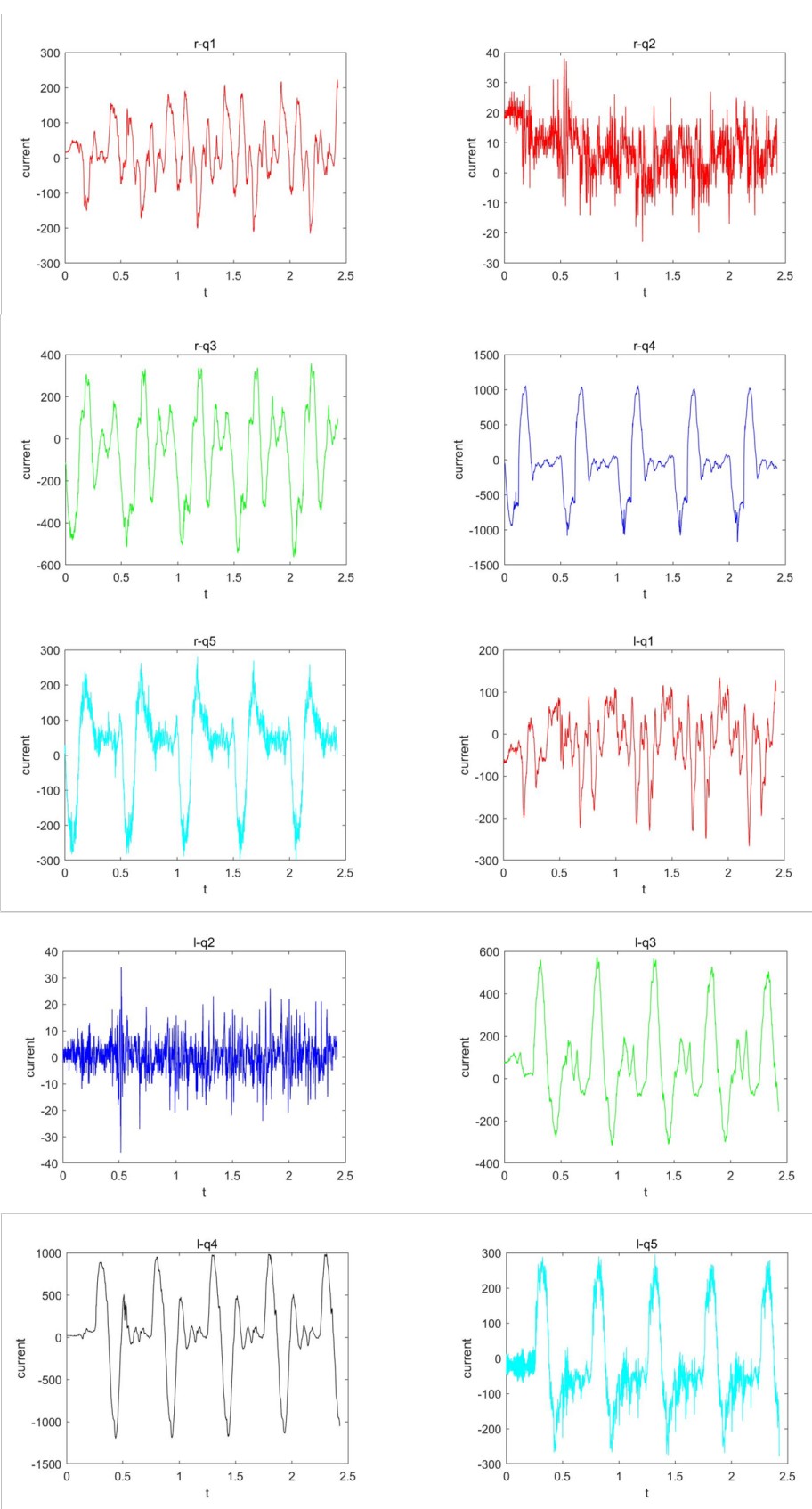

**Figure 11.** Experiment data. When the trapezoidal trajectory speed is 8 mm/MS, this is the motor current data collected by the robot motion. It can be seen from the figure that when the robot moves at the current speed, it approaches the performance limit of the motor, and even has a short-term current overload.

### 4.2.2. Experiments B

The main purpose of Experiment B is to evaluate whether the motor performance can meet the needs of the robot. It is hoped that the experimental results can provide meaningful information for the optimization direction of the second-generation prototype. In the experiment, a T-shaped curve is used. The uniform speed is 3 mm/ms. The following Figures 12 and 13 are the actual motion diagram of the robot and the collected data diagram. It can be clearly seen from the data map that during the ground motion, the data of motor $q_1$ has exceeded the rated current of the motor. This indicates that the performance of the motor $q_1$ cannot meet the needs of the robot. The reason why the current value oaf motor $q_1$ is too large is mainly due to the insufficient performance of the motor. During the experiment, the motor $q_1$ has no input. The center of gravity of the robot is not on the supporting polygon during its movement, and the like opportunity wobbles left and right. This will bring greater load pressure to the motor $q_1$. However, the performance of the motor itself cannot support this heavy load, so the current value exceeds the rated value. At the same time, in the course of the experiment, a large displacement occurs when the motor $q_1$ cannot be supported. Other motors have considerable margin of data from the rated current, but further performance improvements are needed to better meet the requirements.

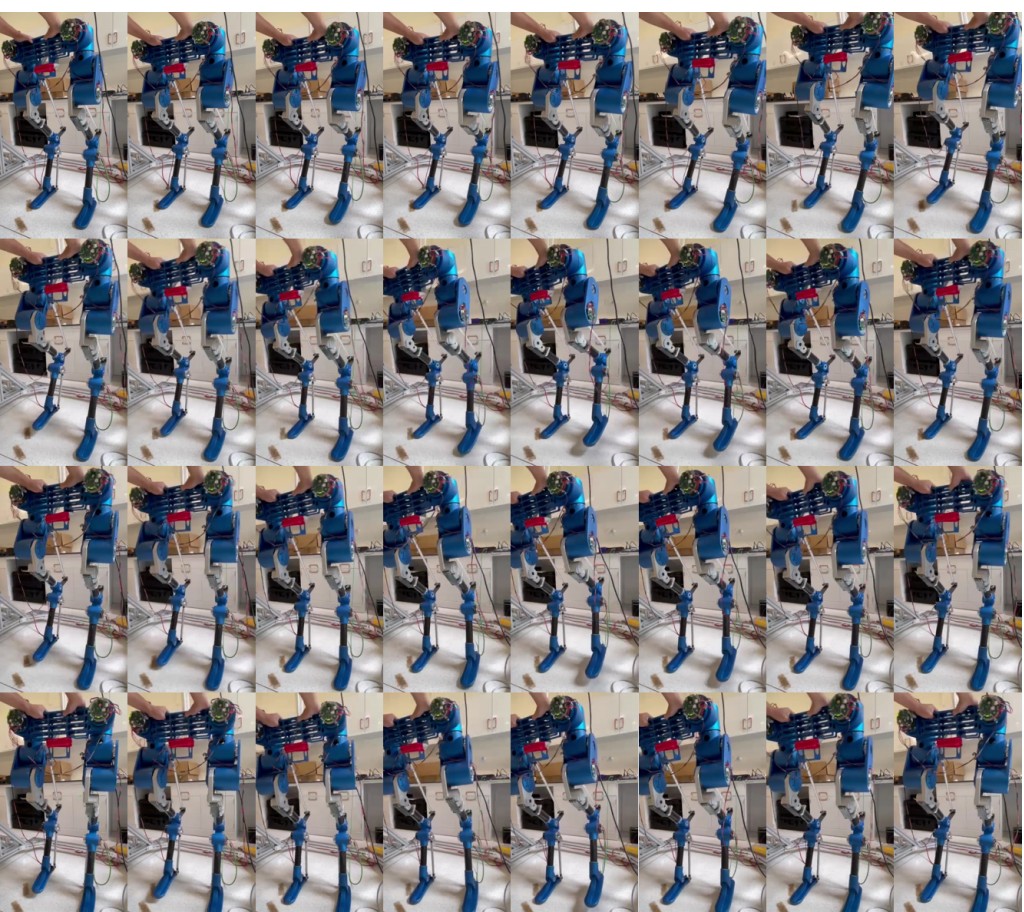

**Figure 12.** The prototype walks step on flat ground.

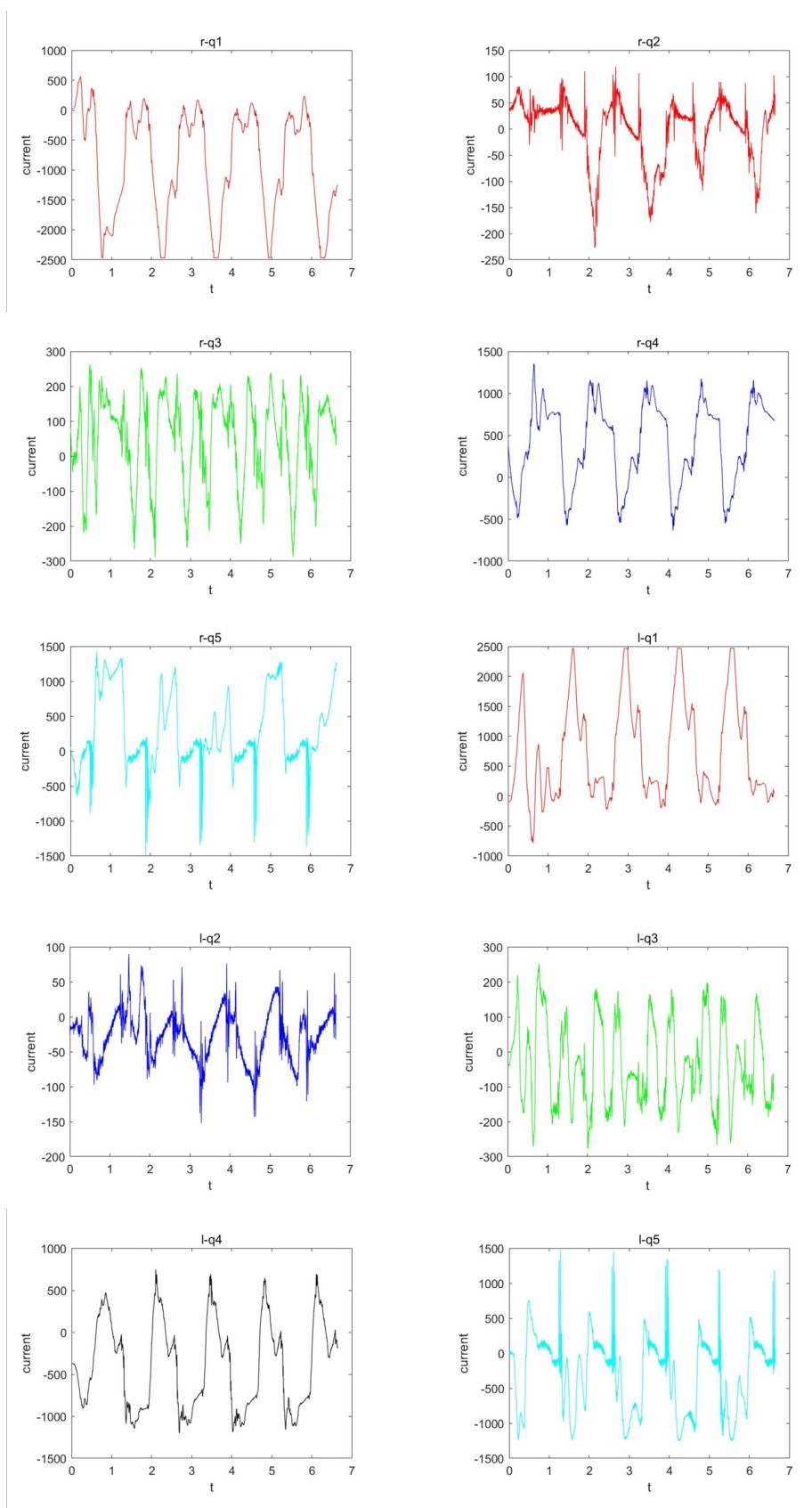

**Figure 13.** Experiment data. When the robot steps at the speed of 3 mm/ms on the ground, the current data collected by the driver. Some motors exceed the rated current during movement, and even greatly exceed the rated current for a long time. Among them, the motor $q_1$ once exceeds more than twice the rated current during movement.

## 5. Conclusions

In this paper, the kinematics of a hybrid mechanism based on spring mass model is modeled, and its forward and inverse kinematics are solved. In the simulation, the kinematic model is verified. Then according to this leg structure, a biped robot prototype is built. In order to test whether the prototype meets the performance requirements of dynamic motion, some performance tests are carried out for the robot prototype. During the test, Some mechanisms are optimized. It is also found that the performance requirements of motor $q_1$ are very high in dynamic motion. Motor $q_1$ does not meet this requirement. Although the performance of other motors indicates that they can meet the corresponding motion requirements, only structural parts and motors are assembled on the prototype body. If it is equipped with batteries, controllers, IMU and other devices, whether the performance of other motors meets the requirements or is unknown. So, it still have a long way to go to get the robot fully moving.

In the future, the prototype needs to be more thoroughly optimized. First, the body is overweight. Its parts, motors and other equipment need to be more thoroughly lightweight. Secondly, the motor performance cannot meet the motion performance requirements. In order to be able to carry devices such as batteries and controllers, the performance of the motor needs to be improved objectively, while the weight should not be too large. The self-design of the motor is necessary. Thirdly, biped robot walking requires a high balance and stability algorithm. The biped robot requires a high center of gravity during its traveling, and it needs to satisfy the ZMP balance and stability criterion at all times. The next generation of robots needs to be improved in algorithm to alleviate the burden on the motor. Finally, the feedback data of the simulation environment is inconsistent with that of the prototype. This requires dynamic parameter identification of the robot to make the simulation consistent with the inertia and other parameters of the prototype. This makes it possible to make the simulated feedback data closer to the prototype.

**Author Contributions:** Conceptualization, J.C. and Y.P.; methodology, J.C. and Y.P.; software, J.C., W.Y. and Y.P.; validation, J.C. and Y.P.; formal analysis, J.C. and Y.P.; investigation, J.C.; resources, Y.P.; data curation, J.C.; writing—original draft preparation, J.C.; writing—review and editing, J.C. and Y.P.; visualization, J.C. and J.Y.; supervision, Y.P.; project administration, J.C. and Y.P. All authors have read and agreed to the published version of the manuscript.

**Funding:** This research was funded by SUSTech Institute of Robotics-LeJu(Shenzhen) Robotics Joint Research Center on Robotics grant number K2033Z054.

**Institutional Review Board Statement:** Not applicable.

**Informed Consent Statement:** Not applicable.

**Data Availability Statement:** If you need, you can find the experiment data at https://github.com/Gabriel-cjj/Biped_rbt, accessed on 16 February 2022.

**Acknowledgments:** This work was supported in part by the Science, Technology and Innovation Commission of Shenzhen Municipality under grant no. ZDSYS20200811143601004.

**Conflicts of Interest:** The authors declare no conflict of interest. And the funders had no role in the design of the study.

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
