# Peer review of "Leg Configuration Analysis and Prototype Design of Biped Robot Based on Spring Mass Model"

_actuators, doi:10.3390/act11030075_

Round 1
Reviewer 1 Report
This manuscript presents the kinematics model of a bionic biped robot, which uses the screw theory to establish the forward and inverse kinematics models. Authors built a prototype and used a step gait to test the model and prototype. This work has potential application for the design and iteration of biped robot prototype. However, this manuscript can be improved considering the following comments:
1.-Abstract section should add the main advantages of the proposed model.
2.-Introduction section should include the main scientific contribution, advantages and limitations of the proposed kinematic model and built prototype with respect to other works reported in the literature.
3.-Authors must check that all the parameters of equations are correctly described. For instance, p and Po from equations (2) and (3).
4.-Equation (1) must be revised.
5.-Captions of all the figures should be improved. These captions must contain more information of the contain of the figures.
6.-Simulation section is very short. This section should add more information and parameters used in the simulation models of the walking of the robot.
7.-Figures 11-14 and 16 have size small. The size of these figures should be increased. In addition, authors should include more discussion of these figures.
8.-Which are the main challenges of the proposed model and built prototype?
9.-Authors should add more recent references between 2019 and 2021.
Author Response
Thanks for your review. I have revised the paper according to your comments.
Point 1: Abstract section should add the main advantages of the proposed model.
Response 1: Without the description of the advantages and disadvantages of the mechanism, the summary can not well show the significance of the work. I have added the main advantages of the proposed model in abstract.s
Point 2 and 8 : Introduction section should include the main scientific contribution, advantages and limitations of the proposed kinematic model and built prototype with respect to other works reported in the literature.
Response 2: In the introduction section, the lack of the main scientific contribution, advantages and limitations of the proposed kinematic model will will lead to my following can not well reflect the importance of work to readers. The description of this part is added in the last paragraph of introduction. As for the construction of other relevant prototypes, this is very meaningful, but it takes a very long time to design and debug. During the epidemic, time is a scarce resource. I hope to have the opportunity to carry out relevant work in the future.
Point 3 and 4:Authors must check that all the parameters of equations are correctly described. For instance, p and Po from equations (2) and (3). Equation (1) must be revised.
Response 3 and 4:This type of error has been checked and modified.
Point 5 and 7:Captions of all the figures should be improved. These captions must contain more information of the contain of the figures. Figures 11-14 and 16 have size small. The size of these figures should be increased. In addition, authors should include more discussion of these figures.
Response 5 and 7:Due to the problem of periodical format, only about two-thirds of the area left for the text and pictures in a page is not conducive to the display of pictures, which also leads to a large proportion of pictures in the content of the article. The pictures and descriptions in this article have been optimized to show the work content more clearly.
Point 6:Simulation section is very short. This section should add more information and parameters used in the simulation models of the walking of the robot.
Response 6:Simulation is a focus, but not the focus of this paper. This part is really relatively simple. At present, relevant descriptions and parameters have been added. Because of the original design problems, the gait of the robot is limited. This one also needs to be solved. It is hoped that more progress can be made in relevant aspects after the successful development of the next generation prototype.
Thank you again for your review. This gives me a lot of very meaningful suggestions. I will learn. I hope to complete more perfect papers in the future。
Reviewer 2 Report
The paper entitled “Leg Configuration Analysis And Prototype Design of Biped
Robot Based on Spring Mass Model” is poorly written although the research presented in is a mature state. Many of the presented results were previously presented without being cited making as for example in the following publication:
https://dl.acm.org/doi/10.1145/3462648.3462652
by this way it in not clear what are the main new contributions of this paper.
The chapters are not balanced, chapter 2 and 3 are too small, probably chapters 2, 3 and 4 made sense just as one chapter.
Some graphs do not have good quality making the paper less comprehensive.
No future or previous work is presented.
The authors write many times in the first person what should be avoided.
The presented research is relevant, but the presented information must be presented more carefully so the readers can understand the evolution of the work and clearly understand the new contributions and the proposed new contributions.
Author Response
Dear reviewer,
Thanks for your review. It helped me a lot.
Point 1:
Robot Based on Spring Mass Model” is poorly written although the research presented in is a mature state. Many of the presented results were previously presented without being cited making as for example in the following publication:
https://dl.acm.org/doi/10.1145/3462648.3462652
by this way it in not clear what are the main new contributions of this paper.
Response 1: This paper does have many similarities with previous papers. But this paper perfects what the previous paper did not complete. For example, firstly, due to the limitation of space and meeting time, the previous papers are not perfect in the inverse kinematics. Secondly, the prototype has been further optimized and changed in structure, so corresponding modifications have been made in this paper.
Point 2: The chapters are not balanced, chapter 2 and 3 are too small, probably chapters 2, 3 and 4 made sense just as one chapter.
Response 2: The separation of these three chapters will indeed appear loose as a whole. I've combined it.
Point 3: Some graphs do not have good quality making the paper less comprehensive.
Response 3: Due to the problem of periodical format, only about two-thirds of the area left for the text and pictures in a page is not conducive to the display of pictures, which also leads to a large proportion of pictures in the content of the article. The pictures and descriptions in this article have been optimized to show the work content more clearly.
Point 4:No future or previous work is presented.
Response 4:The future work mainly focuses on the design and development of prototype and the optimization of solution. However, due to the problem of motor processing, it takes too much time and energy.
Point 5: The authors write many times in the first person what should be avoided.
Response 5: Using too much first person is really not rigorous and too oral. I have revised the relevant rhetoric.
Thank you again for your review. This gives me a lot of very meaningful suggestions. I will learn. I hope to complete more perfect papers in the future.
Kind regards,
Junjie Che
Reviewer 3 Report
The paper presents a Leg Configuration Analysis And Prototype Design of Biped Robot.
The paper is well structured in formal analysis and manufacturing. However to strengthen the soundness of the paper more detailed analysis on the state of art comparing it to the proposed structure have to be done. These paper can be added to the references and cited to help achieve the purpose.
Cafolla, D., Ceccarelli, M. Design and FEM analysis of a novel humanoid torso (2015) Mechanisms and Machine Science, 25, pp. 477-488. DOI: 10.1007/978-3-319-09858-6_45
Russo, M., Cafolla, D., Ceccarelli, M. Development of LARMbot 2, a novel humanoid robot with parallel architectures (2019) Mechanisms and Machine Science, 66, pp. 17-24. DOI: 10.1007/978-3-030-00365-4_3Figure 11 to 14 have to be improved since the yare hard to read both in quality and dimensions.
Author Response
Dear reviewer,
Thanks for your review. It helped me a lot.
Point 1: Cafolla, D., Ceccarelli, M. Design and FEM analysis of a novel humanoid torso (2015) Mechanisms and Machine Science, 25, pp. 477-488. DOI: 10.1007/978-3-319-09858-6_45
Response 1: This is a paper on the control of the upper body of the robot. It has a strong reference value for humanoid robot with upper body. Although my prototype has only the lower part of my body, this article still has a lot to learn. In the future, the upper body will be loaded for the prototype, and this article will be quoted at that time.
Point 2: Russo, M., Cafolla, D., Ceccarelli, M. Development of LARMbot 2, a novel humanoid robot with parallel architectures (2019) Mechanisms and Machine Science, 66, pp. 17-24. DOI: 10.1007/978-3-030-00365-4_3
Response 2: This paper describes a lightweight, low cost solution based on parallel architectures for torso and leg mechanisms The kinematics is constructed and a prototype is built for verification. Its legs adopt a parallel structure. This can enrich the prototype types of introduction in the first draft. Relevant prototypes have been added to the first draft, and this article is cited.
Thank you again for your review. This gives me a lot of very meaningful suggestions. I will learn. I hope to complete more perfect papers in the future.
Kind regards,
Junjie Che
Round 2
Reviewer 1 Report
This second version of manuscript has been improved considering the reviewer's comments.
Author Response
Dear reviewer
Thank you for your review and approval to my paper. It helped me a lot.
Kind regards,
Junjie Che
Reviewer 2 Report
If this paper “perfects what the previous paper did not complete”, the previous paper or papers must be cited, in order to enhance and show clearly which are the new contributions. The remaining points were clarified by the authors.
Author Response
Dear reviewer
Thank you for your review. I have improve considering your comments.
Point: If this paper “perfects what the previous paper did not complete”, the previous paper or papers must be cited, in order to enhance and show clearly which are the new contributions. The remaining points were clarified by the authors.
Response: It is really important to state that the paper is based on the progress of past work. This related content is mainly the second chapter. In third version of manuscript, I have described the relevant progress in this chapter.
Thank you for your help again.
Kind regards,
Junjie Che